# Transcriptional Condensates: Epigenetic Reprogramming and Therapeutic Targets in Hematologic Malignancies

**DOI:** 10.3390/cancers17193148

**Published:** 2025-09-27

**Authors:** Kevin Qiu, Qing Yin, Chongzhi Zang, Jianguo Tao

**Affiliations:** 1Department of Pathology, University of Virginia School of Medicine, Charlottesville, VA 22903, USA; qcx9hm@virginia.edu (K.Q.); jxr8vk@virginia.edu (Q.Y.); 2Department of Genome Sciences, University of Virginia School of Medicine, Charlottesville, VA 22903, USA; zang@virginia.edu

**Keywords:** transcription, transcriptional condensate, epigenetics, targeted therapy, oncogenesis

## Abstract

Emerging evidence supports the notion that transcriptional condensates, clusters of proteins that dynamically form and disassemble based on liquid–liquid phase separation via multivalent interactions, play a key role in the regulation of gene expression. These condensates dynamically interact with gene promoters and enhancers to regulate transcription. They can assemble in response to cellular signals to govern transcriptional efficiency and specificity in normal biologic processes. However, this transcription machinery is among the mechanisms hijacked in cancer, providing a key link into how specific gene alterations can enact broad changes in downstream abnormal gene expression. Transcriptional condensates therefore serve as an Achilles heel to cancers that rely on them. Understanding how to disrupt them may inform the future for potent and specific anti-cancer therapy.

## 1. Introduction

Transcription, the formation of an RNA molecule from a DNA template, is a pivotal step in gene expression. It involves a highly orchestrated interaction network of transcription factors (TFs), transcription initiation and elongation proteins, and chromatin modifiers. Gene transcription is initiated with the binding of RNA polymerase to a target promoter, typically guided by TFs that bind to specific DNA sequences and recruit additional elongation factors alongside RNA polymerase II [1]. These DNA sequences, known as “enhancers”, are therefore key mediators of transcriptional activation despite not typically producing functional RNA themselves [2]. Enhancers can further potentiate transcription by serving as hubs for coactivators, proteins that bind TFs themselves and increase the rate of transcription of a gene or a set of genes [3]. For example, coactivators with general activation domains commonly assist in recruiting additional histone modifiers to reshape nearby chromatin to be more amenable to polymerase binding [4]. Other coactivators induce 3-dimensional conformational changes that permit a TF to co-localize to a DNA enhancer or promoter sequence [5,6,7]. This delicate orchestra of transcription factors, cofactors, chromatin regulators, and transcription machinery interacting with DNA elements dictates the expression of the very genes that govern specific cellular behavior. To enact even broader effects such as dictating cellular identity, super-enhancers (SEs) exist as clusters of enhancers with enrichment of H3K27ac, TFs, and coactivators [8,9]. Because of their broad reach, such regions can be exploited by cancer cells through oncogenes to enhance their pro-survival and proliferative phenotype. SEs therefore underlie the spatiotemporal regulation of oncogenic gene expression and drive progression and therapy resistance [9,10].

Cancers are remarkably dynamic in their ability to trigger specific cellular functions or cellular trajectory states to adapt to their environment. These distinct states are gained or lost by shuttling transcription machinery across a densely populated nuclear space to the right genomic region at the right time. Accumulating evidence now suggests that this coordinated governance is achieved through the formation of transcriptional condensates (TCs), a nuclear subtype of membrane-less microcompartments. These droplet-like structures concentrate proteins and nucleic acids, often forming through liquid–liquid phase separation (LLPS) of proteins that contain intrinsically disordered regions (IDRs) [11,12,13]. Naturally, super-enhancers also rely on the formation of nuclear TC’s to promote activation of transcriptional programs with broad functional consequences, including in the context of cancer aggressiveness and drug resistance [14]. In fact, biomolecular condensates are foci at all levels of gene expression [15,16,17,18] and chromatin organization [19,20,21,22] in both normal and malignant cells. To this end, we summarize the research landscape surrounding how phase-separated condensates form at SEs and how they offer novel insights into mechanisms involved in the epigenetic control of cancer biology, and introduce alternative strategies for treating hematological tumors

## 2. Liquid-Liquid Phase Separation and Aberrant Biomolecular Condensates in Cancer

### 2.1. Liquid-Liquid Phase Separation and Aberrant Biomolecular Condensate Formation

Nearly all cellular functions ranging from genome manipulation to transcription and intracellular signaling rely on the formation of membrane-less compartments to some degree. Liquid–liquid phase separation (LLPS) is a spontaneous and reversible process in which intracellular molecules, such as proteins and nucleic acids, are sequestered from the greater cellular environment and allowed to interact with each other. These membrane-less organelles are known as biomolecular condensates [23]. LLPS is typically triggered when participant molecules exceed a certain concentration threshold, making it thermodynamically favorable for them to unite and interact with one other [24]. Naturally, this threshold is modified by specific pH levels and temperatures, thus making these formations fundamentally subject to the whims of transient physicochemical forces. This property implicates LLPS in regulating a myriad of spatiotemporal cellular functions, including protein–protein interactions at the cell membrane and in the cytoplasmic, signal transduction, and gene expression [25]. Moreover, compartmentalized condensates can still exchange components with the surrounding cytoplasm or nucleoplasm, allowing the cell to introduce elements that further modulate the biological reactions contained within them (Figure 1) [26,27,28,29]. Under normal conditions, cells have a variety of mechanisms to tightly regulate LLPS; however, an increasing number of aberrant LLPS are now being described in many oncogenic processes [29].

The defining characteristic of the protein or nucleic acid molecules involved in phase separation is the existence of multivalent and adhesive low-complexity domains. Condensation-prone proteins frequently exhibit several flavors of this feature: (1) multimerization domains that facilitate protein–protein interactions, (2) nucleic acid binding domains that drive interactions with DNA or RNA, and/or (3) long and flexible intrinsically disordered regions (IDRs). IDRs are of particular interest as they exhibit sequence-intrinsic bias towards conformational heterogeneity, allowing them to create weak multivalent interactions between proteins and nucleic acids [30,31,32]. The large amplitude of conformational fluctuations mediated by IDRs allows for the recruitment of other molecules such as protein partners, nuclear co-activators, and RNA polymerase into the condensates. This makes them important players in the molecular dynamics in LLPS-mediated condensates [33]. In fact, it has been estimated that most cell signaling proteins and cancer-related proteins also have long IDRs [34], including ones involved in uncontrolled cell proliferation, increased cell survival, and therapy resistance [35,36,37].

### 2.2. Nuclear Transcriptional Condensates in Transcriptional Regulation

The initiation of LLPS typically begins in tandem with transcription, canonically with the hyper-phosphorylation of the carboxy-terminal domain (CTD) of RNA polymerase II (RNAPII) [38,39]. This repetitive, unstructured, IDR-containing region allows RNAPII to undergo phase separation at cis-regulatory elements such as promoters, enhancers, and SEs. This would then allow RNAPII to aggregate into condensates, creating microscopic puncta-like compartments. Experiments reducing the length of the CTD decreased Pol II clustering and RNA transcription and vice versa [40].

At the same time, TFs undergo IDR-mediated LLPS either by themselves or by incorporating themselves with co-activators as part of a larger conglomerate. Co-activators such as MED1 and BRD4 also undergo phase separation at enhancers and super-enhancers in normal cells [15,41,42]. Following initial condensate formation, RNAPII will have its CTD phosphorylated by cyclin-dependent kinase 7 and 9 (CDK7/9), allowing it to separate from the initial condensate and initiate a new LLPS instance with elongation factors [15,41,42,43,44]. Following condensate formation, RNAPII will again have its CTD phosphorylated, this time by cyclin-dependent kinase 7 and 9 (CDK7/9), allowing it to separate from the initial condensate and initiate a new LLPS instance with elongation factors [15,43,44]. All interactions can be modified further by physical and chemical forces, which could potentially be manipulated to influence TC formation and dissolution [16,45,46,47].

### 2.3. Nuclear Transcriptional Condensates at SEs Drive Oncogenic Transcription

Enhancers are non-coding cis-regulatory elements that modulate gene expression through the recruitment of transcription factors [48,49]. While enhancers are discrete genomic elements that control one or more gene(s), super-enhancers regulate a large number of genes that define cell identity [50,51,52]. TC formation has been described in both. Hnisz et al. established a phase separation model of SEs and demonstrated that the key co-activators BRD4 and MED1 compartmentalize and at these regions [15,41]. SEs controlling proliferation, plasticity, and immune escape can be hijacked during oncogenesis [9] and play a role in mediating expression of oncogenes such as MYC in hematological tumors [53,54]. Bahr et al. utilized CRISPR-Cas9 genome editing to systematically delete individual enhancer elements within the BRAM super-enhancer controlling MYC expression in AML. Deletion of these “hub” enhancers caused near-complete loss of MYC expression and leukemic cell death, whereas deletion of other nearby elements had minimal effect.

Indeed, the high-density transcriptional apparatuses at SEs may rely on biomolecular condensate formation for their ability to sharply form, self-segregate, and rapidly collapse on demand [43,52,55,56,57,58,59]. Using live-cell super-resolution and light-sheet imaging technologies, Sabari et al. [15] similarly showed that BRD4 and MED1 can phase separate into compartmentalized condensates in SEs and concentrate RNAPII and the Mediator complex to initiate gene expression [15,60]. The authors used multiple complementary approaches to validate the phase separation phenomenon, including demonstrating liquid-like fusion events between condensates, rapid internal molecular rearrangement via FRAP analysis, and sensitivity to 1,6-hexanediol treatment. Importantly, they showed that condensate formation occurred at endogenous protein concentrations and that artificially tethering the IDRs alone to chromatin was sufficient to create transcriptional hot spots [61,62]. Another recent work by Du et al. used live-cell super-resolution imaging approaches to directly observe endogenous transcriptional condensates’ function at SEs to regulate the expression of Sox2 [63], which further supports the phase-separation model.

Transcriptional condensates may rely on the framework of 3D chromatin structures as they incorporate both promoter-bound Pol II and enhancer-bound transcription coactivators [41,43,64,65,66,67,68]. The transcription factor CCCTC-binding factor (CTCF) is well described to induce DNA looping and insulating lineage-specific transcription factors at super-enhancers [69]. Lee et al. demonstrated that CTCF is essential for RNA polymerase II (Pol II)-mediated chromatin interactions and occurs as hyperconnected spatial clusters at super-enhancers [70]. Interestingly, clusters of Pol II, BRD4, and MED1 were found to dissolve upon CTCF depletion but reappeared after it was re-expressed, suggesting that CTCF is necessary to instruct the formation of transcriptional condensates. These data support the notion that CTCF-mediated looping serves as an architectural prerequisite for TC assembly.

## 3. Nuclear Transcription Condensates Driving Transcription Regulation in Hematologic Malignancies

### 3.1. Aberrant LLPS and Biomolecular Condensates in Cancer

Like in physiological LLPS, condensation in cancer occurs when a critical concentration of an oncoprotein reached through either overexpression or enhanced stability of the corresponding oncogene. This can occur from mutations that change to the tertiary/quaternary structure of a protein or create new intrinsically disordered regions. Alternatively, mutations in the chromatin and transcription regulators contained within in these phase-separated TCs also have oncogenic potential [71]. Hematological malignancies have been shown to rely on transcription programs dysregulated exhibiting condensate formation (Table 1). For example, in multiple myeloma (MM), proteasome inhibitors were reported to dissolve condensates and induce chromatin condensation [72] at the c-MYC super-enhancer, leading to downregulation of c-MYC and its targets. In aggressive B cell lymphomas, the translocation of the IgH enhancer to the MYC locus is a crucial oncogenic event that commandeers the constitutively active TC present at the enhancer to MYC (Figure 2) [55,73]. In T cell acute lymphoblastic leukemias, T-ALL, a mutation to a site upstream of TAL1 was described to allow the master TF MYB to congregate in a large super-enhancer that reciprocally drives TAL1 expression [74]. In this case, a single mutation led to condensate formation, rapidly developing into the ectopic formation of an apparatus containing hundreds of transcriptional components with broad regulatory potential [75]. In acute myeloid leukemia (AML), Terlecki-Zaniewicz et al. demonstrated that NUP98-associated TF chimeras form nuclear condensates, induce aberrant chromatin looping, and drive leukemogenic gene expression programs [76,77]. They identified over 200 proteins specifically enriched in NUP98-fusion condensates compared to normal nuclear speckles. ChIP-Seq and Hi-C sequencing confirmed that condensates formed during NUP98-fusion created aberrant chromatin loops at the megabase resolution, bringing distal enhancers within reach of key proto-oncogenes. Small molecule disruption of an IDR present on the fusion protein dissolved these condensates within 30 min and with reversal of pathogenic gene expression. Ahn et al. complemented this work by developing an in vivo model to visualize real-time condensate formation during leukemogenesis and showed that formation of NUP98-HOXA9 fusion condensates preceded gene expression changes by nearly 6–12 h. Using single-molecule tracking, they demonstrated that condensate formation reduced the search time for target genes by approximately 100-fold, providing a kinetic advantage for oncogenic transcription. Their mathematical modeling suggested that this enhanced search efficiency could explain the dominant oncogenic effects of NUP98 fusions even in the presence of wild-type transcription factors. Another example of aberrant LLPS driving tumorigenesis comes from recent work on the histone demethylase UTX (ubiquitously transcribed tetratricopeptide repeat on chromosome X), which plays an essential role in developmental gene regulation and is a primary tumor suppressor [78,79]. Our group determined that the tumor-suppressive effects of UTX phase-separated condensates were unaffected by the selective deletion of most structures but was abolished by deleting the IDR-containing region, thus supporting the notion that the IDR found in UTX was necessary for co-condensing chromatin modifiers at key genomic sites [78,80]. Together, these data highlight the current knowledge about the function and localization of transcription machinery proteins in the context of biomolecular condensates in leukemias and lymphomas. Uncovering how compartmentalization of proteins, RNA, and DNA occurs through biomolecular condensation, therefore, is pivotal to truly understanding the organization and regulation of cellular processes.

### 3.2. Multi-Omics Data Science Strategy to Inform Spatially Clustered Patterns of TF Binding and Phase-Separated TCs at SEs

Advances in principled computational modeling of DNA sequence features has shown that TF concentrations above sharply defined thresholds can drive the formation of localized condensates in order to promote enhancer activity and transcription (Table 2) [75]. However, deciphering the condensation potential across permutations of different TFs in different cell types remains an ongoing effort. Nevertheless, a common theme emerges in the notion that LLPS-mediated condensate formation is a potent mechanism behind SE-driven gene expression, and their activity requires the binding of both cell-type-specific factors and sequence-dependent effectors to drive the formation of localized condensation [16,75,84,85]. Importantly, condensates rely on hydrophobic residues for LLPS to happen, a dependency that Shi et al. exploited when they introduced anti-1,6-HD index of chromatin-associated proteins (AICAP) assay. The aliphatic alcohol 1,6-hexanediol (1,6-HD) disrupts chromatin-adjacent condensates and releases proteins contained within these compartments back into the greater cyto/nucleoplasm. The AICAP value, or concentration of the protein content before/after 1,6-HD treatment, is therefore inversely related to the proclivity of a protein to undergo phase separation and form condensates [86]. The authors used this novel quantification method alongside Hi-MS and Hi-C data to show that regulatory compartment rich in transitory protein/DNA interactions were more sensitive to 1,6-HD.

To push this method forward, we used a data-driven computational approach to explore the connection between genomic TF binding patterns and LLPS affinity [87]. We performed a comprehensive survey of the genomic spatial distribution patterns of cis-regulatory elements using high-quality ChIP-seq binding profiles to evaluate the clustering tendency of actual TF binding sites. We found that many transcription factors exhibited tendencies to cluster beyond their established motifs, particularly at cell-type-specific super-enhancers. Using AICAP, we found that a factor’s propensity to form DNA-binding clusters significantly correlated with its ability to form phase-separated condensates in certain cell types. This novel correlation supports the idea that the clustered pattern of genomic binding and the phase separation capabilities of TFs are both tied to the formation of transcriptional condensates. We also postulated that factors with high cell-type-specific cluster propensities may also have critical oncogenic functions in their corresponding malignancies. Additionally, when we analyzed hundreds of CTCF ChIP-seq datasets and other large-scale genomic data from various human tissues and cancer samples, we identified cancer-specific gains and losses of CTCF binding. We postulated that these binding differences did not always arise from changes in DNA sequence or methylation state, but rather changes in CTCF recruitment by clusters of cancer-specific TFs and cofactors during condensate formation. This work implicates CTCF in maintaining the active chromatin state through transcriptional condensates, particularly during oncogenesis [88]. However, more experiments are needed to functionally validate these theories and find suitable applications for them.

**Table 2 cancers-17-03148-t002:** Experimental and NGS methods utilized in characterizing transcriptional condensates.

Method	Mechanism	Resolution	Advantages/Use Cases
FRAP [89]	Measure molecular dynamics via photobleaching recovery	~250 nm	Quantitative dynamics, live cells
OptoDroplets/OptoIDR [90]	Light-controlled condensate formation	Single condensate	Precise temporal control, reversible
ChIP-seq/CUT&RUN [91]	Map genomic localization of TC components	~200 bp	Genome-wide, quantitative
Hi-C/HiChIP [92,93]	3D chromatin organization, loop detection	1–10 kb	Links structure to function
Live super-resolution [63]	Visualize endogenous condensates	20–50 nm	Native protein levels, dynamics
AICAP assay [86]	Screen for phase-separating proteins	N/A	Proteomics screening
Single-molecule tracking [94]	TF search kinetics and binding	~30 nm	Direct mechanism, absolute measurements

## 4. Transcriptional Condensates as a Novel Vulnerability in Hematologic Malignancies

### 4.1. Targeting the SE Transcription Machinery of Transcriptional Condensates

As we dissect the mechanisms and downstream effects of TC-mediated phase separation in epigenetic regulation in normal and malignant states, we begin to see possible targets emerge as potential therapeutics against aggressive and treatment-resistant cancers (Figure 3, Table 3). One such mechanism would be to inhibit protein constituents of the transcription complexes aberrantly attracted to cancer-related condensates. Some small-molecule inhibitors can bind the active sites of these ligand-able proteins, thus preventing their activity in these TCs. These small molecules include those targeting BRD4, cyclin-dependent kinases (CDK7 and CDK9), and BET. For instance, the BRD4 inhibitor JQ1 exhibits a high binding affinity to the bromodomain pocket and disrupts the interaction between BRD4 and acetylated lysine residues, thereby preventing its recruitment to SE sites [95]. Because BRD4 physically interacts with the Mediator complex, the application of JQ1 can lead to rapid release of Mediator, leading to repressed transcription of downstream MYB target genes in leukemias/lymphomas [96]. JQ1 exposure also led to preferential inhibition of SE-driven MYC transcription in multiple myeloma [55] and B-cell lymphoma [81]. This was shown when leukemias with acquired resistance to JQ1 were found to activate MYC in the absence of BRD4 by substituting β-catenin at the unoccupied BRD4 binding sites [97,98]. Inhibition of the Wnt/β-catenin pathway resensitized the cells to BET inhibitors [97]. Similarly, another report demonstrated that suppression of the PRC2 complex promotes AML resistance to BET inhibitors through the restoration of oncogenic Myc transcription through Wnt recruitment and the activation of a proximal enhancer [98]. Similarly, the covalent CDK7 inhibitor THZ1 suppresses CDK7-dependent phosphorylation and achieves clinical activity in chronic myelogenous leukemia (CML) [56]. THZ1 disrupts the transcription of SE-associated gene XBP1 and showed potent effect against CML stem cells [99]. We and others showed that THZ1 at lower doses downregulates global transcription by specifically decreasing enhancer-driven transcription programs, including MYC and other cancer-specific oncogenic TFs and signaling molecules in aggressive B-cell lymphomas [100]. In our previous work on mantle cell and aggressive large cell lymphoma, super-enhancers gained in response to ibrutinib and venetoclax treatment were specifically disrupted when we inhibited CDK9, providing us an avenue to explore the prevention or negation therapy resistance evolution [83,100].

### 4.2. Disruption of Transcriptional Condensates at Silent Enhancer Sites

Another attractive treatment modality is to simply reverse the condensate formation process. One druggable target is the Hippo pathway, which relies on LLPS-mediated condensation to mediate cancer cell survival and immune evasion via YAP and TAZ upregulation [107,108]. YAP and TAZ condensates concentrate TEAD4, BRD4, MED1, and CDK9 at super-enhancer sites to initiate YAP target gene expression [17,18]. Cai et al. showed that specific dissolution of YAP condensates with verteporfin reversed chromatin-level topological changes associated with oncogenesis within 4 h, demonstrating the dynamic and reversible nature of YAP condensate-mediated genome organization. Perturbation of YAP and TAZ condensates by the anti-HIV drug elvitegravir inhibits the proliferation of tumor cells dependent on YAP target genes, demonstrating that preventing phase-separation of oncogenic condensates is a potent therapeutic strategy [106], potentially alongside established immunotherapies such as anti-PD-1 immunotherapy. Additionally, existing antineoplastic drugs have been shown to selectively concentrate within phase-separated condensates compared to the rest of the cyto/nucleoplasm [109,110]. For example, one group found that MED1 condensates preferentially concentrate the chemotherapy drug cisplatin through two independent mechanisms, as MED1 condensates attract molecules aromatic rings and form at SEs susceptible to platination [104]. Cisplatin treatment induced gradual and specific disruption of MED1 condensates, demonstrating a potential role for platinum drugs to effectively treat tumors that are dependent on MED1 SE-driven oncogene expression. Tamoxifen, another antineoplastic drug used in the treatment of estrogen receptor positive breast cancer, concentrates with ERa in MED1 condensates and competes with estrogen within the sequestered space [104]. In AML, the small molecule BET bromodomain inhibitor JQ1 combines the previously discussed mechanisms, inhibiting condensates formed by BRD4, NPM1, and MED1 [104] while also interfering with its ability to bind to its epigenetic targets [15,104]. Other enticing targets exist in AML, including TCs formed by hydrophobic interactions between NUP98 fusion proteins, the IDRs of the transcriptional coactivator EP300, and NF-kB, leading to histone acetylation at promoters of protooncogenes [111]. Therefore, while EP300-specific inhibitors such as A-485 have already been shown to trigger robust disassembly of EP300-containing condensates [112], they may also have a function in NUP98-fusion cancers by disrupting condensate constituents themselves. Such examples provide strong evidence that the targeted application of small-molecule therapies in condensates not only enhances drug pharmacodynamics of pre-existing antineoplastics but can also be used to specifically interfere with these elusive phase-separated complexes and nullify their epigenetic regulatory activities.

### 4.3. Drugs Directly Interacting and Targeting IDRs

Although IDRs were long thought to be undruggable, they have recently emerged as promising targets for disrupting LLPS in pathogenic condensates. Theoretically, peptides that occupy intermolecular interaction sites could reduce the valency of condensate-forming proteins regardless of the physical mechanisms driving condensate biogenesis, provided that multivalency is a requirement. Unfortunately, viable small molecules and inhibitors are very limited, as IDRs lack the stable secondary and tertiary structures or recognition sequences typically targeted by conventional approaches. Existing small molecules that do bind IDRs generally do so with very low affinity [113,114,115,116,117]. Nonetheless, recent advances in chemical engineering have stepped up to this challenge. For example, the compound PCG, derived from Polygonum cuspidatum Sieb, has been shown to directly bind BRD4’s IDR and effectively convert phase-separated nuclear BRD4 into static, nonfunctional aggregates both in vivo and in vitro [102,118,119]. Wang et al. showed that PCG treatment in AML cell lines reduced BRD4 condensate number by nearly 75% within 6 h and decreased MYC expression by 60%. Crucially, while ablating the BRD4 IDR with PCG showed good efficacy against AML cell lines, it had minimal effects on cells lacking BRD4 amplification. Other compounds, such as IIA4B20, IIA6B17, and mycmycin-1/2, can effectively target the IDRs of the MYC oncoprotein, inhibiting its oncogenic activity and malignant cell transformation [103,120]. Other small molecules that target the MYC IDR and components of the transcription initiation complex TFIID have also been identified [121]. These studies show that polar and repetitive IDRs should be pursued as a new avenue for inhibiting transcriptional condensates.

### 4.4. Degrading Biomolecular Condensate via Proteolysis

Somewhat antithetical to their key roles in RNA and protein synthesis, condensates also serve as molecular intermediates for proteasome function and autophagy, which are necessary for degrading cellular proteins [121,122]. BCL6, a member of the BTB/POZ zinc finger family of transcriptional repressors, is often constitutively expressed in diffuse large B-cell lymphomas (DLBCLs) [123] and is the most commonly altered oncogene in B-cell lymphomas. Sustained expression of BCL6 drives malignant transformation of germinal center B-cells by repressing key repressor genes such as cell cycle checkpoint genes. The small molecule BI-3802 was found to form condensates by directly binding BCL6 dimers through the stabilization of a dimerization interface [82]. These condensates then recruit E3 ubiquitin ligase to initiate proteasomal degradation of the BCL6 aggregates far more efficiently than conventional BCL6 inhibitors.

This degradation process is often enhanced by the presence of chaperone proteins such as Hsp70. These proteins are ubiquitous in the protein-folding process and can assist in refolding stress-denatured proteins, and interfacing with the ubiquitin–proteasome system to degrade terminally misfolded proteins [124,125,126]. In fact, heat-induced aggregation, long thought to be the result of toxic misfolding, has recently been attributed to adaptive chaperone-mediated biomolecular condensation. Substantial in vivo evidence shows that heat-induced biomolecular condensates are major endogenous substrates of molecular chaperones [61,127], which help to modify and prevent condensate formation [127]. A key step in this process is Hsp70 localizing to a diverse host of “client” proteins by the J-domain protein (JDP) family of adapters [128]. Zhang et al. generated the synthetic JDP analog JDM37, capable of binding Hsp70 at the same site as the native JDP [129]. JDM37 stimulated ATPase activity and, intriguingly, dissolved two different condensates [130]. In line with this, the same group also demonstrated that generating a fusion protein containing the DNAJB1 J-domain to the catalytic subunit of protein kinase A disrupts PKA condensation and led to oncogenesis. Inversely, the H33Q mutation in the J-domain abolishes Hsp70 activation and does not affect PKA condensation [131]. JDPs co-localize substrates with Hsp70s through J domain–Hsp70 binding and activate Hsp70 to disperse aggregation and condensation in the targets. Thus, directing condensate proteins to Hsp70s via J-domain disrupts not only nuclear condensation but also the underlying condensate-dependent biological function. It would be reasonable to suggest that homing Hsp70 to specific subcellular locations alongside substrate-specific synthetic JDPs could have broad clinical significance.

## 5. Concluding Remarks and Future Directions

Recent studies suggest that several oncogenic and tumor-suppressive proteins perform their functions within specific membrane-less cellular compartments. Condensates involved in cancer development and therapy resistance are promising targets for treatment of cancers, especially when first-line therapies have failed. Because phase separation is a highly sophisticated biological process, small doses of small molecules, antibodies, or peptide treatments targeting this process could be highly effective in disrupting condensates. Several groups have demonstrated that targeting TF-mediated phase separation is both feasible and effective in suppressing oncogenic transcription, inhibiting tumor growth, and reversing drug resistance [132]. Conversely, because genome stability is also closely associated with LLPS, restoring physiological LLPS may also help slow or prevent tumorigenesis.

Nevertheless, further research into the functional exploration of LLPS in cancer is warranted, as many problems remain unsolved. Much more work is needed to unravel the biological, chemical, and physical interactions between every single molecule and constituent structure within LLPS, as well as understanding how they drive oncogenic functions. Methods need to be developed to model the thermodynamic forces derived from cellular activities such as transcription, post-translational modification, and m6A modification that cooperatively regulate phase separation. Another area requiring further research is the therapeutic window associated with targeting protein condensates. Unlike conventional cytotoxic or more targeted therapies, which have toxicities that can be reasonably minimized by tailoring drugs towards novel mutations or fusions, phase separation is a ubiquitous process in normal and pathological cells. Additionally, condensates are highly transient and, up until recently, challenging to reliably quantify; so, therapies targeting condensates are harder to predict, risk low tumor specificity, and can be highly toxic towards normal cells. It is imperative that condensates identified through FRAP and AICAP be carried out in both malignant and normal cells, and that multi-omics methods such as RNA-Seq and chromatin binding sequencing be employed to further distinguish condensate formation between normal and pathological conditions to maximize therapeutic selectivity.

While understanding the incredibly intricate higher-order interactions between biomolecules such as proteins, RNA, and DNA behind transcriptional condensates may seem herculean, the work conducted so far has made leaps and bounds in terms of deepening our comprehension of cancer biology. Further study of these mechanisms is needed to develop new drugs with improved sensitivity and specificity for malignant cells. The effects of TFs and epigenetic regulators on super-enhancers during tumorigenesis, as well as the impact of these molecules on the electric charge of IDRs and protein complexes, should also be investigated. Overall, we strongly believe that further exploration of these mechanisms may yield a new treatment paradigm for cancers and beyond.

## Figures and Tables

**Figure 1 cancers-17-03148-f001:**
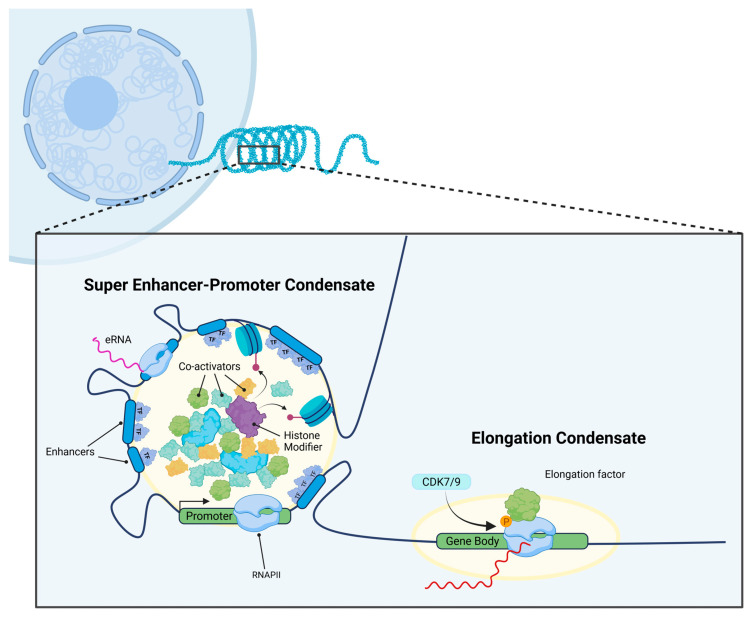
Overview of TCs in the context of the standard transcriptional model, a phase separated region physiochemically distinct from the rest of the nucleosome, allowing for the accumulation and isolation of RNA polymerase, histone modifiers, and transcription factors & co-activators in order to initiate transcription.

**Figure 2 cancers-17-03148-f002:**
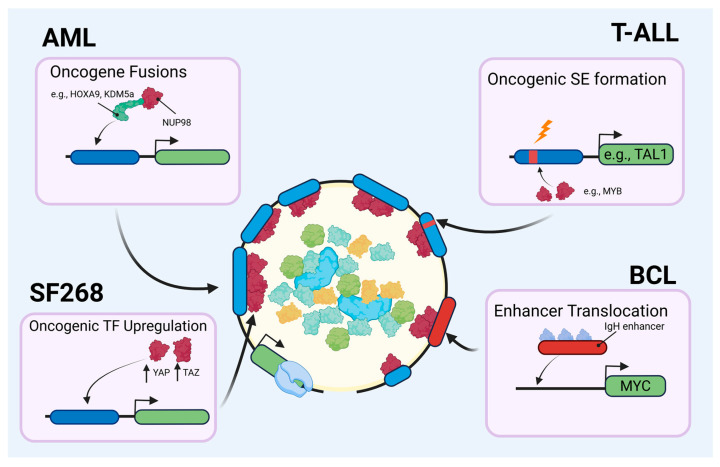
Broad overview of established oncogenic pathways in the context of transcription condensates in the nucleus.

**Figure 3 cancers-17-03148-f003:**
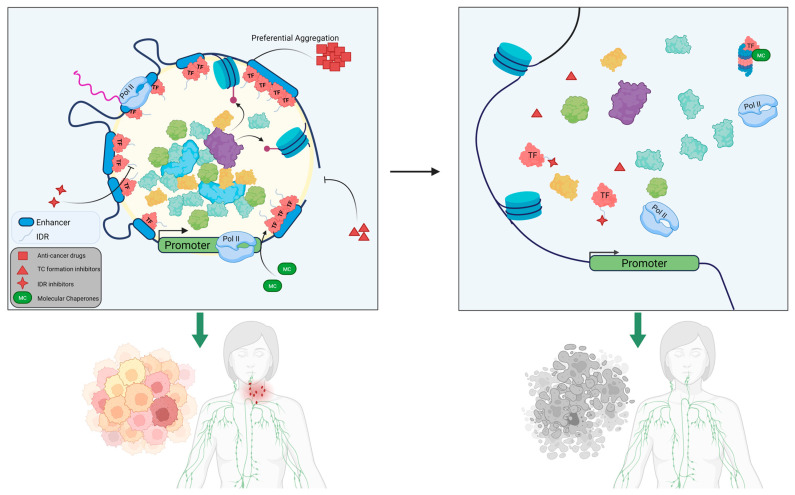
Potential treatment modalities specifically targeting TCs, including small molecule therapies that preferentially accumulate within TCs, drugs that block intrinsically disordered regions (IDRs) or TC components and thus block TC formation, or recruitment of proteasome and chaperone proteins that mediate TC degradation.

**Table 1 cancers-17-03148-t001:** Hematologic malignancies with known alterations in transcriptional condensates.

Malignancy	Prevalence of TC Alterations	Condensate Components
AML [76]	5–11% pediatric, 2–4% adult AML	NUP98 chimeras, EP300, CREBBP, BRD4, MED1
DLBCL [81,82]	BCL6+ in 70%, BRD4+ in 85%	BCL6, BRD4, MED1, p300
Multiple Myeloma [55,72]	MYC dysregulation in >40%	BRD4, CDK9, HDAC3
T-ALL [74]	TAL1 alterations in 60%	MYB, TAL1 SE
CML [56]	CDK7-dependent in all cases	CDK7, BRD4, MED1
MCL [83]	CDK9 addiction post-ibrutinib	CDK9, BRD4, P53

**Table 3 cancers-17-03148-t003:** Therapeutic strategies targeting transcriptional condensates.

Strategy	Anti-TC Mechanism of Action	Drug Examples	Resistance Mechanisms
BET Bromodomain Inhibition [55,97,98]	Displaces BRD4 from acetylated chromatin	JQ1, OTX015, INCB054329	Wnt/β-catenin activation, PRC2 loss
CDK7 Inhibition [56,101]	Blocks Pol II CTD Ser5 phosphorylation	THZ1, SY-1365, CT7001	MYC-independent programs
CDK9 Inhibition [83,100]	Prevents transcription elongation	Dinaciclib, AZD4573	Alternative elongation factors
Direct IDR Targeting [102,103]	Binds IDR motifs, prevents LLPS	PCG, mycmycin-1/2	Unknown
Induced Degradation [82]	Polymerization triggers proteasomal degradation	BI-3802	Proteasome inhibition
Condensate Drug Partitioning [104]	Selective accumulation in TCs	Cisplatin (repurposing), tamoxifen (repurposing)	Condensate dissolution
YAP/TAZ Modulation [17,105,106]	Disrupts Hippo pathway condensates	Verteporfin (FDA approved), elvitegravir (repurposing)	YAP-independent pathways

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
