# Peer review of "Transcriptional Condensates: Epigenetic Reprogramming and Therapeutic Targets in Hematologic Malignancies"

_cancers, 2025, doi:10.3390/cancers17193148_

Round 1
Reviewer 1 Report
Comments and Suggestions for Authors
refer to attachment

requires significant corrections. Professional help can be considered.
Reviewer 2 Report
Comments and Suggestions for Authors
The review of Qiu, Yin, and Tao is comprehensively written, round, and easy to understand. It gives a complete understanding of the role of transcriptional condensates.
Minor things to consider:
Although transcriptional condensates have a role in tumor initiation and progression, there are several other factors needed. Therefore, I would suggest to milden their CENTRAL role in carcinogenesis.
Page 3, line 133: Is there a verb missing from this sentence “… a repetitive, unstructured, low-complexity region that also an IDR”?
The function of TCs is repeated too many times.
Round 2
Reviewer 1 Report
Comments and Suggestions for Authors
no further comments